# Uremic Toxins and Protein-Bound Therapeutics in AKI and CKD: Up-to-Date Evidence

**DOI:** 10.3390/toxins14010008

**Published:** 2021-12-23

**Authors:** Jia-Huang Chen, Chih-Kang Chiang

**Affiliations:** 1Graduate Institute of Toxicology, College of Medicine, National Taiwan University, Taipei 100233, Taiwan; f04447010@ntu.edu.tw; 2Department of Internal Medicine, College of Medicine, National Taiwan University, Taipei 100233, Taiwan; 3Department of Integrated Diagnostics & Therapeutics, National Taiwan University Hospital, Taipei 100225, Taiwan; 4Center for Biotechnology, National Taiwan University, Taipei 10617, Taiwan

**Keywords:** acute kidney injury, protein-bound uremic toxins, indoxyl sulfate, p-cresyl sulfate

## Abstract

Uremic toxins are defined as harmful metabolites that accumulate in the human body of patients whose renal function declines, especially chronic kidney disease (CKD) patients. Growing evidence demonstrates the deteriorating effect of uremic toxins on CKD progression and CKD-related complications, and removing uremic toxins in CKD has become the conventional treatment in the clinic. However, studies rarely pay attention to uremic toxin clearance in the early stage of acute kidney injury (AKI) to prevent progression to CKD despite increasing reports demonstrating that uremic toxins are correlated with the severity of injury or mortality. This review highlights the current evidence of uremic toxin accumulation in AKI and the therapeutic value to prevent CKD progression specific to protein-bound uremic toxins (PBUTs).

## 1. Introduction

### 1.1. The Updated ADQI Consensus for AKI, AKD, and CKD

End-stage renal disease (ESRD) has had a high disease burden recently, which might be because the kidney itself is a vulnerable organ that easily suffers from numerous pathogenic insults, including nephrotoxic agents, hypoperfusion, metabolic disturbances, uremic toxins retention, and natural aging-related processes. Once an acute kidney injury (AKI) is encountered, the insults frequently lead to subsequence chronic kidney disease (CKD) progression despite a temporary recovery from AKI [1]. On the other hand, cumulative pieces of evidence also suggest that CKD acts as a predisposing factor for newly developed AKI [2]. In order to develop the consensus for AKI, acute kidney disease (AKD), and CKD, Acute Dialysis Quality Initiative (ADQI)16 announces the expert’s meetings and makes a new statement (Figure 1) [3].AKI can be demonstrated as a continuum, while initial kidney damage can lead to later renal injury and eventually lead to CKD. For the definition of AKI, it encounters an abrupt reduction in renal function occurring over 7 days or less. For CKD, it is defined by the persistence of kidney disease for a period of >90 days. Between the two clinical situations, there is a transition status which is named AKD. It describes acute or subacute damage and/or loss of kidney function for a duration of between 7 and 90 days after exposure to an AKI initiating insults. Based on clinical observation, recovery from AKI within 48 h of the initiating event typically heralds rapid recovery from AKI. For patients with pre-existing CKD, the AKI event also can be superimposed on CKD, with AKD existing on a background of CKD. AKD patients with pre-existing CKD have shown a high risk of kidney progression and future ESRD development [4].

### 1.2. AKI Increases the Long-Term Risk of Progressive CKD

The development of AKI is associated with increased morbidity, length of hospital stay, higher hospital costs, economic burden, and increased mortality [5,6]. Even small changes in serum creatinine (sCr) of 0.3–0.5 mg/dL are associated with increased long-term mortality [7]. Recent awareness of the prospect that a portion of patients suffering from AKI might progress to CKD or ESRD requiring maintenance dialysis, with the incidence rate of 4.9 per 100 patients/year [8,9]. According to the 2015 United States Renal Data System (USRDS) report, CKD status changes significantly in the year following an AKI hospitalization. Among patients without baseline CKD, nearly 30% are reclassified as having some degree of CKD, including 0.20% being declared ESRD [10]. Whether the patient progresses to CKD after recovery from AKI (also termed as AKI to CKD transition) and the precise mechanisms of AKI to CKD transition are still under exploration. The possible explanation of AKI to CKD transition may be that the maladaptive repair of either irreversible or sustained cellular injury, which further induces glomerulosclerosis, tubulointerstitial fibrosis, or vascular rarefaction. After suffering from AKI, uremic solutes such as the small molecule uremic toxin sCr and blood urea nitrogen (BUN) are increased and have been suggested as an indicator for kidney injury. However, it was gradually recognized that these tests were neither specific nor sensitive in several clinical practices [11,12]. Furthermore, recent studies have revealed that the level of protein-bound uremic toxins (PBUTs) such as indoxyl sulfate (IS) and p-cresyl sulfate (PCS) in AKI is associated with mortality rate and progress to CKD [13,14]. Therefore, we will systemically review the current literature by searching the terms: AKI, CKD, AKI to CKD, and protein-bound uremic toxins on PubMed in regard to the impact of uremic toxins in AKI-to-CKD transition to provide an alternative concept for the early removal of uremic toxin in the early stage of AKI for therapy.

## 2. Etiology and Pathophysiological Roles of Uremic Toxin

Uremic toxins are metabolites generated from daily food intake and their excretion into the urine through glomerular filtration or active transport by the renal proximal epithelial cell [15]. The accumulation of uremic toxins is characteristic of CKD development and renal function decline [16]. The European Uremic Toxin Work Group (EUTox) documented that uremic toxins are classified into three catalogs based on their physicochemical characteristic: water-soluble small molecule (<500 Da), middle molecule (>500 Da), and protein-bound uremic toxin [15]. Water-soluble small molecule uremic toxins are hydrophilic and easily penetrate from the glomerular barrier and are eliminated. Creatinine and urea are the most representative of two small-molecule uremic toxins and are widely used as surrogate markers for identifying renal function in the clinic [17]. Middle molecule uremic toxins, such as β2-Microglobulin, FGF23, and pro-inflammatory cytokines IL-1β, IL-6, have evolved mainly to be synonymous with peptides and proteins that accumulate in uremia [18]. The PBUTs, IS and PCS are two of the notorious uremic toxins, which are derived from the amino acid tryptophan and tyrosine, respectively, and are metabolised in the liver separately [19]. Due to their protein-bound characteristic, very little is filtrated through the glomerular barrier and they have a low clearance rate in patients undergoing hemodialysis or peritoneal dialysis [20]. Therefore, they have more severe adverse effects in patients with renal function impairment, and most studies are focus on exploring their deteriorating effects in CKD, cardiovascular disease, and uremic sarcopenia progression [13,21,22,23].

## 3. The Retention of PBUTs in Kidney Disease Models

There are many kidney disease models in uremic toxin research, whether in AKI, CKD, or AKI to CKD. Here, we listed the commonly used animal model in those studies. For AKI studies: Bilateral ischemia-reperfusion injury, cisplatin, and contrast-induced kidney injury model; For CKD studies: 5/6 nephrectomy, adenine diet, and folic acid injection; For AKI to CKD studies: UIRI, unilateral two-stage IRI. Other models such as aristolochic acid and unilateral ureteral obstruction (UUO) would be applied in AKI or CKD model based on investigator’s experimental design [24]. According to disease definition, both AKI and CKD models would cause impaired renal function with increasing BUN and sCr. Moreover, the renal function of the UIRI model would return to baseline within several days in the AKI model, whereas the CKD model lasted. Of note, UIRI model would not induce obvious BUN and sCr upregulation due to the compensatory effect of the contralateral kidney [25]. However, the injured kidney caused intensive AKI-related gene expression-havcr1 and lcn2. Even after 2–16 weeks of UIRI insult, the injured kidney formed unresolved fibrosis development and inflammatory-related genes expression, which were well-recognized features in CKD [26]. Regarding unilateral two-stage IRI, Wei et al. developed a novel AKI to CKD model through 2 weeks UIRI induction, then removed the contralateral intact kidney. This model presented a progressive development of AKI to CKD transition, including sustained renal function dysfunction after nephrectomy and deleterious effects in renal fibrosis [27]. In contrast to CKD, there are seldom studies in the field of PBUTs in AKI. AKI insult causing uremic toxin retention comes from a decline in glomerular filtration rate (GFR), leading to increased small molecule uremic toxins that cannot be eliminated through the glomerular filtrate and this alters the metabolic profiles which promote uremic toxin precursor formation. In the following section, we list the possible reason for PBUTs accumulation after AKI from many aspects, including secretory function decline, metabolic alteration, and precursor synthesis.

### 3.1. Loss of Secretory Function in Renal Tubular Epithelial Cells after AKI

Acute tubular necrosis is the common consequence of AKI [28]. With the loss of epithelial cells in renal tubules, the secretory function of kidneys are dramatically decreased after AKI due to the loss of OAT1/3 expression in proximal tubular cells [29]. The OATs superfamily are the important transporters expressed in RTECs and have a role in substrates reabsorption and elimination [30]. Among them, OAT1/3 are responsible for PBUTs transport from circulation to filtration by the tubular epitheliums and excretion. It has been found that IS accumulates in an in vivo AKI model [29,31]. Different to BUN and sCr levels which return to normal levels in the recovery stage of AKI [25], furthermore, our current work found that accumulation of the secreted PBUT is sustained without an increase in BUN and sCr after 10 days of unilateral ischemia-reperfusion injury mice model [32]. Therefore, we thought UIRI model would be a suitable model to investigate the roles of PBUTs in AKI to CKD transition.

### 3.2. AKI Alert Gut Microbiota Composition Leading to PBUTs Synthesis

The gut microbiota plays a crucial role in the host immune homeostasis and influence extraintestinal biologic functions. The concept of the kidney-gut axis has recently aroused more attention in the field of AKI and CKD research. Yang et al. demonstrated that ischemia reperfusion-induced AKI provokes intestinal dysbiosis, increasing *Enterobacteriaceae* and *Escherichia coli*, and decreasing *Lactobacilli Ruminococacceae* [33]. Gryp et al. reveal that a higher abundance of *Enterobacteriaceae* and *Escherichia coli* was found in CKD patients’ fecal matter [34], and are also responsible for PBUTs and the synthesis of precursors- p-cresol and indole [35,36]. Furthermore, the decreased levels of *Lactobacillales* are associated with CKD development, inflammation, short-chain fatty acid (SCFA) deficiency, and loss of intestinal permeability [37,38]. SCFAs are the metabolites of gut microbiota and serve as the energy source for intestinal epithelial cells, immune modulation, and strengthening intestinal integrity [38,39]. Since gut microbiota and its microenvironment govern PBUTs production, immune regulation, and kidney disease progression, the research about the management of gut bacterial in AKI and CKD is blooming and their therapeutic efficacy is worthy of further exploration [40,41,42].

## 4. The Harmful Effect of PBUTs Accumulation

### 4.1. Potential Mechanism of Uremic Toxin Retention Involved in AKI to CKD Progression

The potential mechanism of IS-induced aggravated AKI to CKD progression is well known, involving increased TGF-β signaling, oxidative stress, inflammatory response, ER stress, and senescence [43,44,45,46,47]. IS induces ROS generation and tubular cell DNA damage leading to tubular cell cycle arrest in the G2M phase. Prolongation of the arrested G2M phase in tubular cells is also recognized as a feature of the maladaptive repair process, which results in AKI to CKD transition [48]. Myofibroblasts play a crucial role in producing excessive extracellular matrix and can be derived from the activation of renal interstitial fibroblasts, perivascular fibroblasts and pericytes, myelin protein zero-Cre (P0-Cre) lineage-labeled fibroblasts contributing to renal fibrosis in animal models [49,50,51]. During exposure to PBUTs, fibroblasts are directly activated by IS or PCS stimulation or indirectly by tubular secreted TGF-β then leading to myofibroblast activation and extracellular matrix deposition [52,53,54]. Chen et al. indicated that oral gavage of the IS precursor-indole accelerated AKI to CKD progression in an ER stress-dependent manner [32]. Furthermore, several studies support the concept that AKI insults promote premature aging development [55,56]. It is worth noticing that the anti-aging gene klotho was found to have a low expression level in CKD tissue [57,58,59]. The underlying mechanism was revealed by Sun et al., in that IS and PCS accumulation increased DNA methyltransferases expression, which contributes to klotho gene hypermethylation and suppresses klotho protein expression [57,60]. Therefore, it is proposed that PBUTs would be the potential risk factors for accelerated senescence after AKI.

### 4.2. PBUTs Retention Increased the Risk of Disease Progression in Multi-Organ Dysfunction

In addition to previous studies that show that AKI is correlated with a higher risk of adverse outcomes [61,62], current research has indicated that PBUTs accumulated in the circulation in AKI patients leads to a systemic impact on multiple organs, including cardiovascular disease and lung injury [63,64]. Wu et al. demonstrated that serum IS levels were elevated in AKI patients and Shen et al. also found in UIRI mice model [65,66]. Accumulated IS impaired endothelial progenitor cell (EPC) function through oxidative stress leading to EPCs senescence, inhibition of proliferation, and apoptosis [65]. In addition to endothelial cell dysfunction during PBUTs exposure, elevated IS and PCS levels are associated with cardiovascular calcification in CKD patients [67]. Therefore, it is assumed that AKI is also a possible risk factor for development of cardiovascular calcification. In the relation to the association between PBUT and lung injury after AKI, Yabuuchi et al. performed bilateral nephrectomy-induced IS accumulation in rat results in acute lung injury through a decrease in aquaporin-5 expression, an essential transporter in maintaining lung homeostasis [68,69].

## 5. Diagnostic Value of PBUTs in AKI to CKD Progression

sCr and BUN levels are surrogate markers currently used to diagnose AKI [17]. However, as defined by serum creatinine accumulation, AKI may not truly reflect tubular injury because of the difference in uremic toxins clearance route, and the absence of changes in sCr does not assure the absence of tubular injury. Moreover, the poor specificity of sCr could be elevated due to dietary intake or drug-induced side effects resulting in the tubular secretion of creatinine [70]. Therefore, it was widely recognized that these markers were neither specific nor sensitive in our clinical practice [71]. Therefore, discovering another reliable renal function evaluation biomarker is an emerging issue for a better clinical prognosis.

## 6. The Arising Concept of PBUTs Removal after AKI

### 6.1. PBUTs Adsorbent

Of extrarenal origin, dysfunction of target organs, including heart, lung, liver, skeletal muscle, or gastrointestinal tract leads to a metabolic imbalance by increased generation, change of metabolism, decreased excretion, or enhanced intestinal absorption of uremic toxins into organ compartments. Therefore, it is reasonable to assume that uremic toxins accumulation in AKI are similar to those in the CKD spectrum but might occur at an accelerated rate. However, the impact of uremic toxins on renal progression after AKI insult remained unknown and deserved further research. Recently, the treatment of uremic toxin adsorbent AST-120 (Kremezin) in CKD patients has been approved in Japan, Korea, and the Philippines [72]. AST-120 is a spherical adsorptive carbon particle with numerous pores able to remove certain acidic and basic organic compounds, especially for protein-bound uremic toxins: IS and PCS. The precursor of IS, indole is absorbed in the intestines by AST-120, which decreases the formation of IS from liver metabolism. To date, studies support the effects of AST-120 on renal outcomes and delay the initiation of dialysis in patients with advanced stage CKD. Despite the positive result to retard CKD progression in several small clinical trials, there are still no reports that unequivocally demonstrate the improvement of hard renal endpoints [73]. Currently, the concept of uremic toxins’ early elimination has been proposed in several studies. Chen et al. found that oral intake of AST-120 can alleviate IS accumulation-induced renal fibrosis and ER stress in a two-stage IRI model [32]. Shen et al. demonstrated that the AST-120 administration in a mouse IRI model improved cardiac dysfunction through IS-mediated NF-κB/ICAM-1 proinflammatory signaling and apoptosis (Table 1) [74].

### 6.2. Probiotic Supplements

The health benefits of probiotics supplements have caught worldwide attention [75], especially for the CKD population. There are many clinical trials that show promising outcomes in the prevention of renal function decline and PBUTs accumulation in CKD cohorts [76,77,78]. However, there is only one ongoing phase 3 clinical study to investigate the effect of probiotic supplements in sepsis-induced AKI patients (NCT03877081) and one research article that applied probiotic supplements in an AKI rat model. Lee et al. demonstrated that supplementation with the probiotic-*Lactobacillus salivarius*: BP121 can protect against cisplatin-induced AKI through ameliorating renal function parameters BUN and sCr, and reducing serum IS and PCS formation in circulation (Table 1) [79]. Moreover, BP121 supplement also inhibited cisplatin-induced inflammation, oxidative stress, apoptosis, and modulation of the gut environment.

### 6.3. Albumin Binding Displacer

The feature of protein binding of uremic toxins makes it difficult for them to pass through hemodialysis membranes [80]. To solve this problem, Madero et al. demonstrated a novel strategy to enhance PBUTs’ removal efficacy during hemodialysis through pretreatment with ibuprofen—an albumin binding competitor (Table 1) [81]. The idea comes from studies that indicate that the albumin-binding site of IS and PCS can be replaced by ibuprofen in vitro and ex vivo [82,83]. It has to be noticed that this not necessarily a safe option as NSAIDs should be avoided in ESRD patients on dialysis. However it is an interesting proposition that should be further explored safer albumin binding competitors. Recentlyly, Li et al. applied salvianolic acids, an albumin-binding competitor extracted from *Salvia miltiorrhiza* (Danshen) [84], liberating the free form of IS and PCS from albumin binding, which provided superior PBUTs clearance efficacy than conventional hemodialysis [85]. Moreover, salvianolic acids are well known for its anti-oxidative activity. Therefore, it would be a suitable and safe PBUTs binding displacer for further study. Despite the therapeutic benefit of PBUTs binding displacers in CKD still to be determined, a growing number of PBUTs binding displacer studies prove promising therapeutic strategies for PBUTs removal.

### 6.4. SULT Inhibitors

Sulfotransferase (SULT) is a hepatic enzyme responsible for the xenobiotic detoxification process, known as the phase II conjugating procedure that transfers a sulfonate group to the substrate to generate hydrophilic products for urinary excretion [86]. The hepatic phase II biotransformation also plays a critical role in IS and PCS formation in the liver [87]. Therefore, Saito et al. identified the SULT inhibitors—Quercetin, Meclofenamate, and Resveratrol, which were phytochemical polyphenols extracted from Chinese medicine to prevent kidney injury IS, PCS formation in an AKI rat model (Table 1) [88,89,90].

**Table 1 toxins-14-00008-t001:** Summarized the therapeutic strategies for preventing uremic toxin accumulation in AKI stage.

Intervention	Mechanism of Action	Model	Ref
AST-120	Adsorbent	Rat-Cisplatin/IRI	[91,92]/[88]
AST-120	Adsorbent	Mice-IRI	[32,74]
*Lactobacillus salivarius*-BP121	Probiotics	Rat-Cisplatin	[79]
Ibuprofen	Albumin binding displacer	Hemodialysis patients	[81]
Quercetin/Meclofenamate/Resveratrol	SULT inhibitors	Rat-Cisplatin/IRI/IRI	[89]/[90]/[88]
Cobalt chloride	HIF-α Stabilizer	Rat-IRI	[29]

IRI: ischemia-reperfusion injury.

## 7. Conclusions

AKI can demonstrate a continuum, while initial renal injury can lead to kidney damage, and eventually result in CKD and future ESRD development [4]. Since AKI is a risk factor for CKD development and once it progresses to CKD stage, it is hardly reversible. Therefore, eliminating the risk factors for kidney disease development is ongoing in many studies. Moreover, the concept of the deteriorating effect of PBUTs has not been limited to the CKD scenario. Studies in both in animal models and AKI patients have been identified with increased serum IS and PCS levels. Furthermore, several potential therapeutic strategies have been identified that target the removal of PBUTs precursors or prevent their synthesis in the early stage of AKI (Figure 2). However, there is still a need for more evidence to support the benefit of early removal of PBUTs to accelerate relevant clinical trial progression in this cohort.

## Figures and Tables

**Figure 1 toxins-14-00008-f001:**
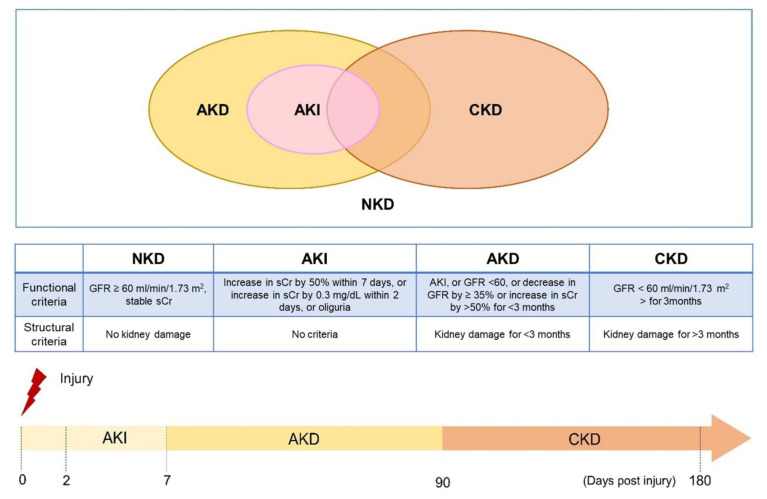
Definitions of kidney disease and the progression timeline (The concepts of the figure were modified from Acute Disease Quality Initiative (ADQI) 16 Workgroup).

**Figure 2 toxins-14-00008-f002:**
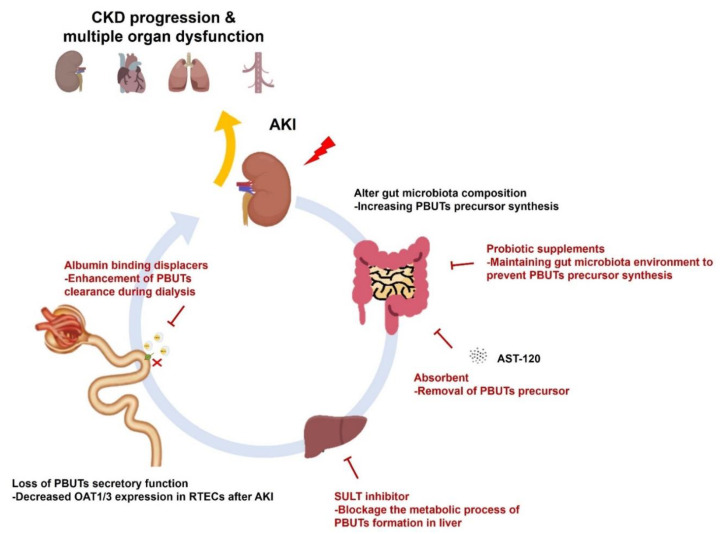
Summary of underlying mechanisms of PBUTs accumulation, adverse outcomes, and potential therapeutic interventions. After suffering from AKI, acute tubular necrosis occurs in RTECs leading to loss of PBUTs transporters OAT1/3 expression and impairment of secretory function. PBUTs precursor-producing bacteria were also found to be increased after AKI and cause indole and p-cresol synthesis. Sustaining PBUTs accumulation in the circulation accelerates the risk of CKD, heart failure, lung injury, and vascular disease progression. To prevent PBUTs accumulation-induced adverse outcomes, developments and interventions such as the oral charcoal adsorbent, probiotic supplements, and SULT inhibitor administration have been investigated in in vivo studies.

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
