# Peer review of "Uremic Toxins and Protein-Bound Therapeutics in AKI and CKD: Up-to-Date Evidence"

_toxins, 2021, doi:10.3390/toxins14010008_

Round 1
Reviewer 1 Report
Comments To Impact of Protein-Bound Uremic Toxins in AKI to CKD: Up-to-date
Evidence
The article is interesting and deals with a major health problem.
DEFECTS TO CORRECT:
TITLE: Where it says: "AKI to CKD"
It should say: "Acute Kidney Injury (AKI) and Chhronic Kidney Disease (CKD)."
ABSTRACT:
Throughout the abstract, the full expression of the abbreviations must be clearly exposed. It is reduced and admits those clarifications, which are necessary.
BODY OF THE ARTICLE:
It remains to state the methodology used for the searches to make this review article.
The terms searched and in which databases must be specified in a Methods section.
Table 1: the meaning of each abbreviation must be specified again in the footer of the table.

Author Response
Reviewer #1
The article is interesting and deals with a major health problem.
DEFECTS TO CORRECT:
TITLE: Where it says: "AKI to CKD"
It should say: "Acute Kidney Injury (AKI) and Chronic Kidney Disease (CKD)."
Response: We appreciate the reviewer’s point out the defects of the review. We would like to highlight the concept that AKI can demonstrate as a continuum leading to AKI to CKD transition. Therefore, we added chapter 1.1. The Updated ADQI Consensus for AKI, AKD, and CKD to enforce the statement (Line 21-41).
ABSTRACT:
Throughout the abstract, the full expression of the abbreviations must be clearly exposed. It is reduced and admits those clarifications, which are necessary.
Response: Thanks for the reviewer’s kindly suggestion, we have added the full expression of the abbreviations in the abstract.
BODY OF THE ARTICLE:
It remains to state the methodology used for the searches to make this review article.
The terms searched and in which databases must be specified in a Methods section.
Table 1: the meaning of each abbreviation must be specified again in the footer of the table.
Response: We appreciate the reviewer’s invaluable suggestion. We have added The terms that we searched on PubMed (Line 63-67). And the meaning of the abbreviations has been added in the footer of table1.
Reviewer 2 Report
The authors provide comprehensive review of PBUTs in light of AKI to CKD progression. Although the hypothesis of PTUC accumulation in CKD progression is probably worthwhile to pursue further, the review concludes in haste with a listing of interventions that are available to help remove PBUTs and a short concluding remarks. The review is already informative and worthwhile publishing, yet a little more exploration into this aspect could make the manuscript much more valuable.
Author Response
Reviewer #2
The authors provide comprehensive review of PBUTs in light of AKI to CKD progression. Although the hypothesis of PTUC accumulation in CKD progression is probably worthwhile to pursue further, the review concludes in haste with a listing of interventions that are available to help remove PBUTs and a short concluding remarks. The review is already informative and worthwhile publishing, yet a little more exploration into this aspect could make the manuscript much more valuable.
Response: We are thankful for the reviewer’s appreciation to this review, and we will add some content followed by reviewer’s suggestions to make the review more valuable.
Reviewer 3 Report
This work intends to review an interesting topic; however, the manuscript needs major language editing to improve understanding. The structure of the manuscript should also be reconsidered. Guided by the title I expected to find a thorough description of the impact of Protein-Bound Uremic Toxins in the development of CKD after an episode of AKI, nevertheless this issue is described very briefly in the manuscript without giving a detailed overview of the topic. Additionally, throughout the manuscript, the authors are presenting different results without offering a deeper analysis of the implication of these data in clinical practice. Moreover, in vitro and in vivo results are presented in a mixed and confused manner without maintaining a rational sequence.
I believe the topic that the authors intent to review is important in the nephrology field, however, major changes need to be done on the manuscript in order to make it suitable for publication.
Author Response
Rewiever #3
This work intends to review an interesting topic; however, the manuscript needs major language editing to improve understanding. The structure of the manuscript should also be reconsidered. Guided by the title I expected to find a thorough description of the impact of Protein-Bound Uremic Toxins in the development of CKD after an episode of AKI, nevertheless this issue is described very briefly in the manuscript without giving a detailed overview of the topic. Additionally, throughout the manuscript, the authors are presenting different results without offering a deeper analysis of the implication of these data in clinical practice. Moreover, in vitro and in vivo results are presented in a mixed and confused manner without maintaining a rational sequence.
I believe the topic that the authors intent to review is important in the nephrology field, however, major changes need to be done on the manuscript in order to make it suitable for publication.
Response: We appreciate the reviewer point out the defect of this review. We have edited the sentence of the review throughout to make the article more readable. Furthermore, we also modified the content of the review to make the article closer to the reivewer’s anticipation.